# Leaf Plasmodesmata Respond Differently to TMV, ToBRFV and TYLCV Infection

**DOI:** 10.3390/plants10071442

**Published:** 2021-07-14

**Authors:** Yaarit Kutsher, Dalia Evenor, Eduard Belausov, Moshe Lapidot, Moshe Reuveni

**Affiliations:** Plant Science Institute, ARO, Volcani Center, 68 Hamakabim Rd, P.O. Box 15159, Rishon LeZion 7528809, Israel; yaarit@volcani.agri.gov.il (Y.K.); vhevenor@volcani.agri.gov.il (D.E.); eddy@volcani.agri.gov.il (E.B.); lapidotm@agri.gov.il (M.L.)

**Keywords:** plant viruses, plasmodesmata, TYLCV, TMV, ToBRFV, tobacco mosaic virus, tomato yellow leaf curl virus, tomato brown rugose fruit virus

## Abstract

Macromolecule and cytosolic signal distribution throughout the plant employs a unique cellular and intracellular mechanism called plasmodesmata (PD). Plant viruses spread throughout plants via PD using their movement proteins (MPs). Viral MPs induce changes in plasmodesmata’s structure and alter their ability to move macromolecule and cytosolic signals. The developmental distribution of a family member of proteins termed plasmodesmata located proteins number 5 (PDLP5) conjugated to GFP (PDLP5-GFP) is described here. The GFP enables the visual localization of PDLP5 in the cell via confocal microscopy. We observed that PDLP5-GFP protein is present in seed protein bodies and immediately after seed imbibition in the plasma membrane. The effect of three different plant viruses, the tobacco mosaic virus (TMV), tomato brown rugose fruit virus (ToBRFV, tobamoviruses), and tomato yellow leaf curl virus (TYLCV, begomoviruses), on PDLP5-GFP accumulation at the plasmodesmata was tested. In tobacco leaf, TMV and ToBRFV increased PDLP5-GFP amount at the plasmodesmata of cell types compared to control. However, there was no statistically significant difference in tomato leaf. On the other hand, TYLCV decreased PDLP5-GFP quantity in plasmodesmata in all tomato leaf cells compared to control, without any significant effect on plasmodesmata in tobacco leaf cells.

## 1. Introduction

The dispersal of macromolecule and cytosolic signals throughout the plant depends on utilizing a unique intracellular apparatus called plasmodesmata (PD). A single plasmodesma is a membrane-coated channel that traverses the cell walls, enabling the transport of molecules and communication between plant cells. The plasmodesmata’s outer membrane is part of the cell’s plasma membrane (PM), and the inner section is part of the endoplasmic reticulum (ER) network. Neighboring plant cells are therefore forming an intracellular domain that connects them directly. Although cell walls are permeable to small soluble proteins and other molecules, plasmodesmata allow direct, regulated, symplastic communication between the cells [1]. Plasmodesmata are tubular, with a central membranous component, the desmotubule, a continuation of the ER lumen, with protein bridges to the plasma membrane that link the PD’s external borders. Macromolecules move between cells through the plasmodesmata within the desmotubule or in the gaps between the plasma membrane and the ER membrane in the plasmodesmata and can be mimicked artificially by selectively forming membrane pores [2].

Plant viruses spread cell-to-cell throughout the plant via PD [3]. Viruses encode movement proteins (MPs) to facilitate their flux through the plasmodesmata channel [4,5]. Tobamoviruses like tobacco mosaic virus (TMV) and tomato brown rugose fruit virus (ToBRFV) have a MP that in the case of TMV was shown to affect plasmodesmata [4,5] While in begomoviruses like tomato yellow leaf curl virus (TYLCV), which is used in this study, an MP was not demonstrated. These viruses are ranked among the top ten plant viruses with research and economic importance [6,7]. These viruses differ in their basic structure and biology. Tobamoviruses genome is composed of a single linear RNA molecule, has no insect vector but infects the plant by physical contact, is found throughout the plant in many cell types, including the mesophyll, and is a model system to study ribonucleoprotein trafficking between cells [7]. TYLCV is phloem located, whitefly-transmitted, and has a genome composed of a single circular DNA molecule [7].

The majority of viral MPs produce minor changes in plasmodesmata’s overall structure [8]. However, some MPs assemble into tubules that alter the plasmodesmata structure and displace the desmotubule inside the plasmodesmata, leaving a tubular plasma membrane that transports virus particles [9,10,11]. Thus, the interaction of plant viruses with plasmodesmata is a possible target for engineering resistance to these pathogens [12,13]. Embryo development depends on functional plasmodesmata [14,15,16], leading to the conclusion that the desiccated seed (and the embryo within) should have plasmodesmata.

A family of proteins termed plasmodesmata located proteins (PDLPs) was identified in *Arabidopsis*. These membrane-bound proteins coat the interior of plasmodesmata and aid plant viruses to traverse from cell to cell [17]. Den-Hollander et al. [18] showed that PDLP1 physically binds viral MPs *in planta* but not in protoplasts and facilitates the formation of tubular channels for viral movement. It was also shown that PDLP5 acts as an inhibitor of macromolecules trafficking via the plasmodesmata [19]. PDLP5 is localized at the central region of plasmodesmata channels in *Arabidopsis* and is associate with pit fields [19]. As a regulator of plasmodesmata, PDLP5 is also essential for conferring enhanced innate immunity against bacterial pathogens in a salicylic acid-dependent manner in *Arabidopsis* [19].

This study investigated the developmental distribution of PDLP5-GFP and the effect plant viruses may have on PDLP5-GFP distribution in tobacco and tomato leaves. Our results show a difference between tomato and tobacco plasmodesmatal response to TMV, ToBRFV, and TYLCV using PDLP5-GFP as a marker protein of plasmodesmata.

## 2. Results

### 2.1. AtPDLP5 Is Present in Plasmodesmata

Expression of AtPDLP5-GFP in tobacco cells shows that it is associated with the cell wall (Figure 1a) with pit fields (Figure 1b,c) and in the guard cell wall that is shared with the epidermis cell (Figure 1c), similarly to its distribution *Arabidopsis* [19]. The expression pattern of AtPDLP5-GFP in tomato was the same as tobacco (Figure 1d–f).

### 2.2. PDLP5 in Seed Germination

We examined the presence of AtPDLP5-GFP in the seed of tobacco and tomato (Figure 2). After one day of dehydration, we looked at seeds to allow easy peeling of the seed coat to view GFP fluorescence. AtPDLP5-GFP protein is present in all seed cells from the root cap to cotyledons (Figure 2). The protein is present in granules (probably protein granules present in the dry seed) (Figure 2). There is no AtPDLP5-GFP presence in the cell walls between the cells or membranes (Figure 2).

Once water hydrates the seed, AtPDLP5 seems to move from storage bodies to the cell wall. An alternative explanation is that atPDLP5-GFP is degraded in the granule bodies, re-synthesized, and moved to the cell membrane. Figure 3 shows a tobacco seed where hydration of the peripheral cell layers is completed, and AtPDLP5-GFP is present in the cell wall. In contrast, in the innermost cells, AtPDPL5-GFP is retained in storage bodies (Figure 3). This process co-occurs in the cotyledons and root tip (Figure 3a,c) while the seed is hydrated. Once at the cell wall, AtPDLP5-GFP organizes into distinct plasmodesmata locations (Figure 3c).

Seven days into the germination process of both tobacco and tomato seeds, while only the radical emerges from the seed, both cotyledons and the meristem already divide and grow, and AtPDLP5-GFP is present in all visible cells at the cell wall (Figure 4).

AtPDLP5-GFP is expressed with the 35S promotor and should be present in all cells. We observed that AtPDLP5-GFP is localized in the cell walls of all cells examined in tobacco and tomato plants. One interesting observation is the large plasmodesmata field at the base of trichomes and hair cells in both tobacco and tomato leaves (Figure 4 and Figure 5c). AtPDLP5-GFP is localized to the of the radical epidermis (Figure 5a); pit fields in leaf vain cells (Figure 5b); the base of the trichomes and between the trichome cells (Figure 5c); leaf epidermis cells (Figure 5d); very young tomato fruit pericarp cells (Figure 5e) and pistil cells (Figure 5f).

### 2.3. Effect of Plant Viruses on PDLP5 Deposition

While PDLP5-GFP plasmodesmata are visible in the four major cell types in the leaf (epidermis cell, hair cell, guard cell, and mesophyll cell), the recruitment of PDLP5-GFP protein plasmodesmata as expressed by GFP intensity differs between cell types and plant type. We assume that GFP intensity is the same under the same excitation/emission and temperature of analysis. In tomato and tobacco leaves, PDLP5-GFP is more intense in epidermis cells. In tobacco, fluorescence is slightest in guard cells, while hair cells and mesophyll cells show an equal and in-between fluorescence intensity compared to tobacco epidermis cells (Figure 6). In tomato leaf, PDLP5-GFP shows the lowest GFP intensity in hair and guard cells (Figure 6), while mesophyll cells in tomato leaf show an in-between GFP intensity and epidermis cells show the highest GFP intensity (Figure 6). Epidermis cells seem to be the most sensitive cells to virus effect as their fluorescence intensity is highest, and thus changes are easily detected.

We examined the effect of different plant viruses on the recruitment of PDLP5-GFP to plasmodesmata of tobacco and tomato after infection. Both plant types were inoculated with TYLCV and TMV. The effect on PDLP5-GFP was examined 30 days post-inoculum when symptoms were visible (Figure 7). We also compared the effect of TMV to ToBRFV infection on PDLP5-GFP fluorescence and chlorophyll fluorescence in epidermis cells (as markers for all cells) in non-infected and control tomato plants (Figure 8).

TMV increases the deposition of PDLP5-GFP in tobacco cells (Figure 7a) significantly, while in tomato cells, there was a minor but constant increase in PDLP5-GFP recruitment that was not statistically significant (excluding for hair cells, Figure 7b). On the other hand, TYLCV caused a decrease in PDLP5-GFP recruitment in tomato cells, except for hair cells of the tomato where TYLCV did not have an effect (Figure 7b). TYLCV did not affect PDLP5-GFP recruitment in tobacco cells except for guard cells that showed a significant increase in PDLP5-GFP, unlike other leaf cells (Figure 7a). ToBRFV treated tomato plants show a similar phenotype as TMV treated tomato in both chlorophyll fluorescence decreases and PDLP5-GFP increases in epidermis cells (Figure 8); other cell types show the same qualitative response as epidermis cells, indicating that different tobamoviruses induce a similar effect on the plasmodesmata.

## 3. Discussion

Plasmodesmata are present in the developing embryo and are essential to its development [14,15,20]. Here we show no plasmodesmata in the dry seed embryo. Instead, a plasmodesmatal protein—PDLP5-GFP—is present as part of the storage proteins in the dry seed. As the seed imbibes water, PDLP5-GFP moves from the storage granules to the plasma membrane and then to the plasmodesmata. Another explanation is that PDLP5-GFP disintegrated in the protein granules and re-synthesized, and deposited in the plasma membrane/cell wall area. Although no fluorescence is present in the cell wall and plasmodesmata in dry seeds, these proteins were shown to be essential to seed germination and development [14,15,20]. We assume that a simple explanation is that the plasmodesmata proteins are moved from storage and transferred to the cell wall on imbibition days before radical elongation is observed and germination occurs. PDLP5-GFP is present in all tissues and seems to be localized to the plasma membrane/cell wall area. PDLP5-GFP decorates both membranes of the plasmodesmata (ER and plasma membrane, see Appendix A). The fluorescence emission resolves to a hollow tube with a solid center very similar to textbook drawings of plasmodesmata. However, while the plasmodesmata opening is depicted as a bagel with the desmotubule in the center, we observed that the desmotubule could move aside, making the gap between the outer plasma membrane and the inner ER membrane roomier on one side (Appendix A). The asymmetric opening could be seen sporadically in EM pictures of plasmodesmata [21,22]. Another possibility is that the off-center location of the desmotubule is an artifact. However, this asymmetry in the desmotubule was detected in other systems by different methods [21,22].

Plant viruses spread throughout the plant via plasmodesmata [3], utilizing their MPs to facilitate the passage in the plasmodesmatal channel [4,5]. TMV, ToBRFV, and TYLCV are disease-causing viruses [23] that are very different in structure [6,7] and nucleic acid composition affecting tomato and tobacco plasmodesmata. TMV viral infection involves plasmodesmata via the virus MP [12,24] and increases the size exclusion limit of plasmodesmata [25]. PDLP5-GFP amount, as indicated by the rise in fluorescence intensity of GFP, increases in plasmodesmata of both tobacco and tomato leaf cells following TMV infection and ToBRFV infection in tomato leaf cells. Both tobacco and tomato plants showed the telltale symptoms of TMV infection. It seems that TMV affects tomato and tobacco similarly by mobilization of plasmodesmata located proteins such as PDLP5-GFP to the plasmalemma and ER membranes that are part of the plasmodesmata. However, in tomato guard, epidermis, and mesophyll cells, the PDLP5-GFP increase was not significantly different, although higher than the control. This observation could be explained the minor effect of TMV on the tomato line we used that showed minor symptoms of TMV. ToBRFV that belongs to the same virus genus as TMV (Tobamoviruses), has a similar but more severe effect on both chlorophyll and GFP fluorescence in epidermis cells, pointing to the general mechanism by which tobamoviruses affect plant cells. This is the first time that ToBRFV is shown to affect plasmodesmata. The PDLP5-GFP increase in fluorescence upon TMV infection and the fact that PDLP5-GFP fluorescence in tomato leaf hair cells does not respond to TYLCV infection can be explained by a large plasmodesmata field present at the base of the hair or trichome cells. The high-density plasmodesmata field at the base of the hair or trichome cells may have different properties than regular PD pits.

TYLCV, a phloem-restricted virus, decreases PDLP5-GFP deposition in plasmodesmata in tomato leaf cells. However, in tobacco, TYLCV did not affect the amount of PDLP5-GFP in the plasmodesmata. The impact of TYLCV seems to be carried all over the leaf, both in tobacco and tomato. However, the virus is considered phloem restricted. This overall effect of TYLCV indicating that although the virus particles were only observed in the phloem, their effect is sensed throughout the tomato leaf cells. It is unclear how TYLCV affects all the leaf cells without being there. However, the observation that PDLP5-GFP deposition in the plasmodesmata is affected by TYLCV may hint that plasmodesmatal communication is involved in TYLCV action. We are trying to transform tomato plants with MP from ToBRFV and TYLCV to analyze the interaction with PDLP5-GFP by crossing the two transgenic plants.

One of the main symptoms of both tobacco and TYLCV is chlorophyll and chloroplast degradation [26], which manifests as yellowing in TYLCV infected plants and mosaic pattern of light and dark green in TMV and ToBRFV infected plants. However, the opposite effect of the two virus types on the recruitment of PDLP5-GFP indicates that the visual symptoms of chlorophyll reduction are not related to the changes that occur in plasmodesmatal proteins or structures. Thus, there seems to be no link between viral effects on plasmodesmata and the effect these viruses have on chloroplast function [26].

## 4. Methods

### 4.1. Virus Maintenance and Whitefly Rearing

Here we show that the Israeli isolate of TYLCV (GenBank Acc. No. X15656) was maintained in tomato (line R13) in an insect-proof greenhouse. Whitefly (*Bemisia tabaci*, biotype B) colonies were reared on cotton (*Gossypium hirsutum* L.) plants grown in muslin-covered cages maintained inside an insect-proof greenhouse. TMV (U1 strain) was maintained on tomato line Rehovot 13 (line R13), and ToBRFV was propagated on tomato line LA3310 (Money Maker with Tm2-2) and kept in an insect-proof greenhouse [12,27].

### 4.2. Plant Material

Transgenic tomato plants (cv Moneymaker) and SR1 tobacco plants were generated following transformation with a binary plasmid containing an *Arabidopsis* 35S::PDLP5 gene fused to GFP that was a gift from Dr. Jung-Youn Lee (University of Delaware, Newark, Delaware, USA). The transformation and regeneration of transgenic tomato and tobacco plants were according to our protocols [28,29]. Plants homozygous to the 35S::AtPDLP5-GFP were used as plant material.

### 4.3. TYLCV and Tobamoviruses Inoculation

Plants were inoculated with TYLCV using clip-cages as described before [30]. Adult whiteflies were allowed for a 48-h acquisition access period (AAP) on TYLCV-infected tomato source plants. Following the AAP, 50 whiteflies were placed in a clip cage. Then one clip cage was attached to the second leaf from the apex of each tomato or tobacco test plant (two- to the three-true-leaf stage). Whiteflies were allowed for a 48-h inoculation access period (IAP) on the tomato test plants. Following the IAP, the clip cages were removed, and plants were treated with imidacloprid (Confidor, Bayer, Leverkusen, Germany) [31,32]. Control plants were treated with whiteflies without TYLCV (non-viruliferous). Plants were maintained in an insect-proof greenhouse at 26–32 °C before analysis at 16 days post-inoculation [31,32]. Infected tobacco plants were verified by PCR analysis due to lack of symptoms (Appendix A).

Test plants infected with TMV or ToBRFV were inoculated mechanically: young leaves of inoculated tomato plants were ground in mortar and pastel and diluted in inoculation buffer (20 mM phosphate, pH 7.4). The leaf extract was applied gently to leaves of tomato and/or tobacco test plants using carborundum as an abrasive [12,27]. After inoculation, the leaves were rinsed with water, and plants were kept in a greenhouse.

### 4.4. TYLCV Detection in Plants

TYLCV DNA was detected using a PCR reaction after DNA extraction. DNA was extracted from plant leaves. TYLCV primers were: TYF = 5′-GCTGATCTGCCATCGATTTG-3′ and TYR = 5′-GGTTCTTCGACCTGGTATC-3′. The PCR was carried out on a Corbett Rotor-Gene 6000 (Qiagen, Düesseldorf, Germany) as follows: 35 cycles of 95 °C 10 s, 60 °C 15 s, and 72 °C 20 s in 20 μL volume included one μL of plant DNA, ten μL of Readymix Kit (Hy labs, Rehovot, Israel), and 1 µL of each primer (0.125 μM) and 7 µL DD sterile water.

### 4.5. Microscopic and Chlorophyll Analysis of Leaf Tissue

Samples were excised from leaf number 7 above the inoculated leaf (leaf number 0 from the top of the plant is the inoculated site) [33]. At TYLCV, TMV, or ToBRFV inoculation time, leaf number 7 from the infection site was already initiated as a small, microscopic leaflet. Leaf number 7 just grew and expanded during the 16 days of symptom development [33]. The parameters that were analyzed per cell type were GFP intensity per area of plasmodesmata. Each treatment was done on five plants twice. Pictures were taken from leaf number 7 from each plant. At least eighteen cells from each cell type were analyzed in each tobacco or tomato leaf. Altogether, each cell type had 300 measurements for each treatment. For tomatoes, there were 410 measurements for each treatment. Leaves for analysis were taken from TYLCV or ToBRFV or TMV-infected tomato plants that showed symptoms, and TYLCV infected tobacco plants were taken after PCR analysis to verify the virus presence (Appendix A).

Images of 35S::PDLP5-GFP were taken with the abaxial side facing the objective, allowing the best visualization of fluorescent signals on the cell wall between adjacent leaf cells [33].

GFP was excited at 488 nm with an argon laser and visualized between 500 and 530 nm. A Leica (Wetzlar, Germany) SP8/LAS X laser scanning confocal microscope was used to observe fluorescently labeled cells. The pictures were analyzed using ImageJ V.1.37 (Wayne Rasband, National Institutes of Health, Beltsville, Meryland, USA(. ImageJ functions analyzed fluorescence intensity.

The picture represents a 3D reconstruction (using Imaris software) of confocal Images collected through image series (Z stacks) that were analyzed using the Bitplane Imaris software Version 8.0.1 (Bitplane A.G., Zurich, Switzerland) as described by Teper-Bamnolker [34].

## Figures and Tables

**Figure 1 plants-10-01442-f001:**
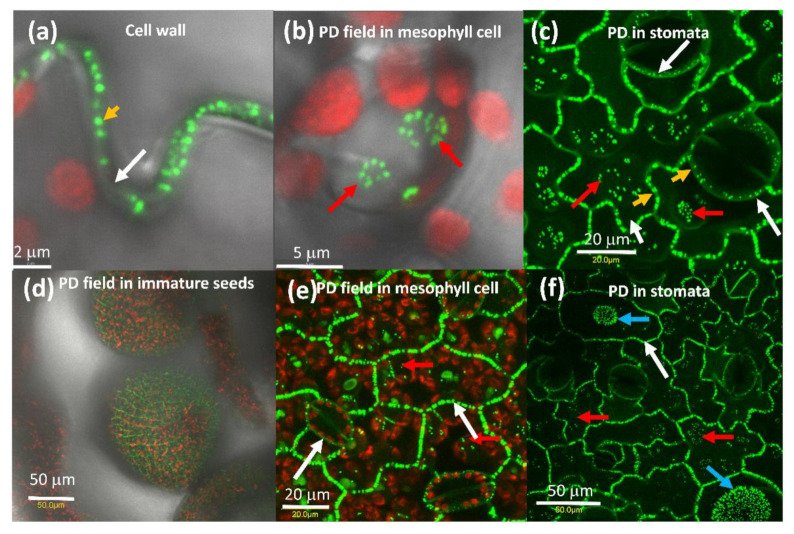
Distribution of PDLP5-GFP tobacco leaf cells. (**a**) Close-up of the tobacco cell wall between adjacent epidermis cells, (**b**) close-up of tobacco mesophyll cell pit field, the red globes are chloroplasts. (**c**) Close-up of stomata in tobacco leaf, (**d**) PD in immature tomato seeds, (**e**) tomato mesophyll cells, (**f**) stomata, and the base of trichomes in tomato leaf. A white arrow marks the cell wall, red arrow pit fields, yellow arrow points to a single PD, and a blue arrow points to pit fields at the base of trichomes. The bar shows the distance in µm.

**Figure 2 plants-10-01442-f002:**
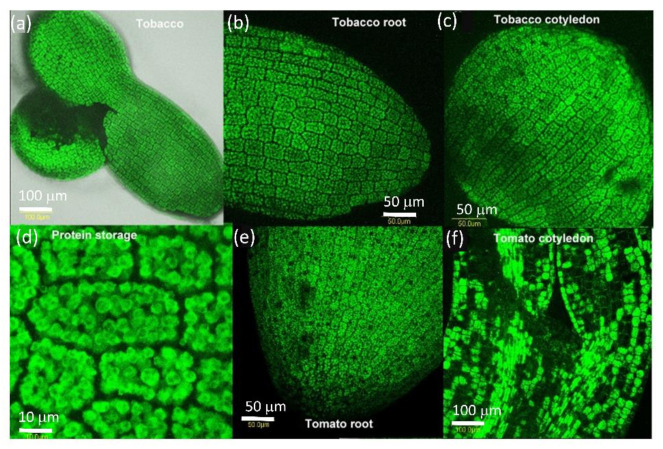
Distribution of PDLP5-GFP in dry seeds of tomato and tobacco. (**a**) Whole dry seed of tobacco, (**b**) close-up of dry tobacco root, (**c**) close-up of dry tobacco cotyledon, (**d**) close-up of dry storage bodies in tobacco root, (**e**) close-up of dry tomato root, (**f**) close-up of dry tomato cotyledon. The bar shows the distance in µm.

**Figure 3 plants-10-01442-f003:**
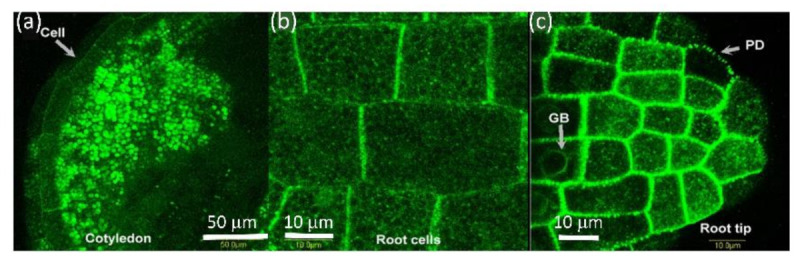
Distribution of PDLP5-GFP in imbibed seeds of tobacco. (**a**) Cotyledon-showing disappearance of PDLP5-GFP from outer cell layer after two days of imbibition. Bar = 50 µm (**b**) close-up of root cells in four days imbibed tobacco seed showing the presence of PDLP5-GFP in the cell wall. Bar = 10 µm (**c**) close-up of root tip cells in two days imbibed tobacco seed. Bar = 10 µm. The bar shows the distance in µm. GB = granular body; PD = plasmodesmata.

**Figure 4 plants-10-01442-f004:**
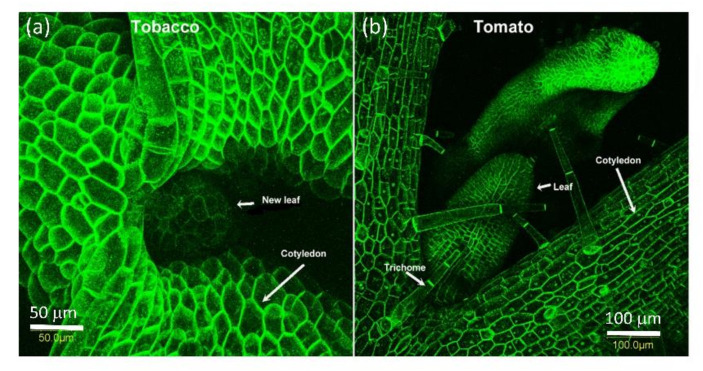
PDLP5-GFP is present in all surface cells of tobacco and tomato 7-day seedlings. The bar shows the distance in µm. (**a**) Meristematic area of 7 days old tobacco seedling where the new leaf emerge between the cotyledons; (**b**) Meristematic area of 10 days old tomato seedling where the new leaves emerge between the cotyledons.

**Figure 5 plants-10-01442-f005:**
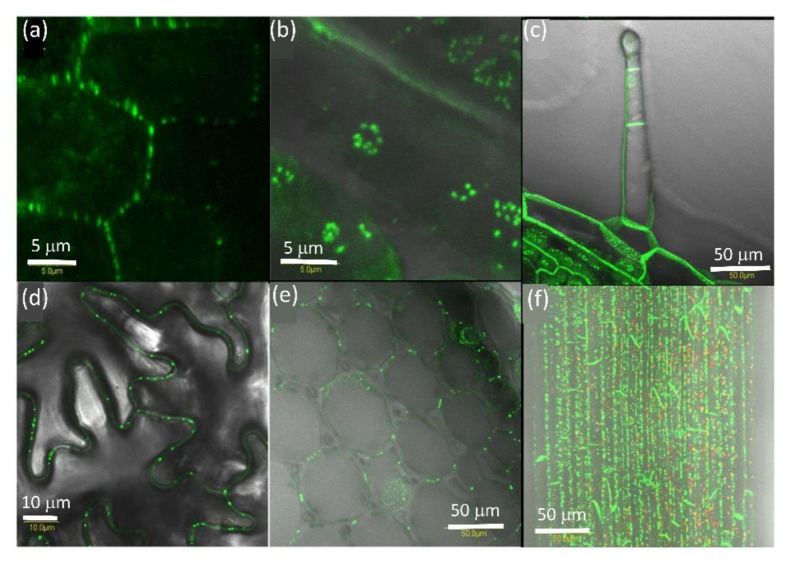
AtPDLP5-GFP is localized to the cell wall of all tobacco and tomato cells. (**a**) Root radical epidermis; (**b**) pit fields in leaf vain cells; (**c**) trichomes; (**d**) leaf epidermis cells; (**e**) very young tomato fruit pericarp cells; and (**f**) tomato pistil cells. The bar shows the distance in µm.

**Figure 6 plants-10-01442-f006:**
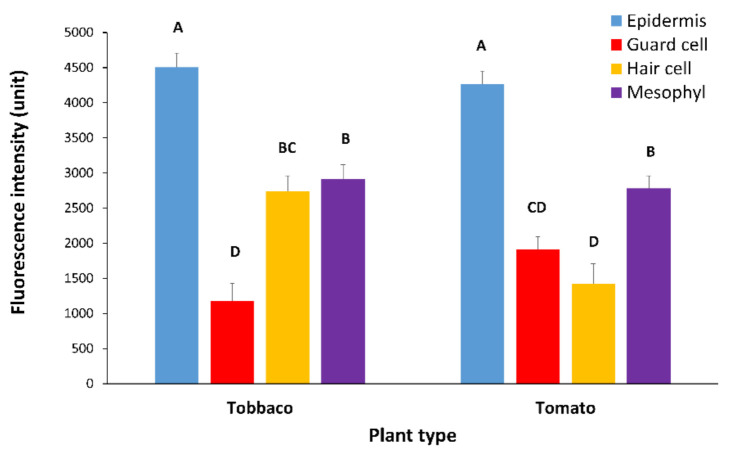
Fluorescence intensity of PDLP5-GFP protein in various cells in leaves of tomato and tobacco plants. Tomato and tobacco plants were transformed with 35S::PDLP5-GFP and were viewed in confocal microscopy. The intensity of GFP fluorescence in the plasmodesmata decorated with GFP per µm of the cell wall was measured per cell type. In each graph, means (± SE) are shown. Different letters above the data columns mean that differences between treatments are statistically significant (*p*{f} <  0.001). Each genotype was tested separately.

**Figure 7 plants-10-01442-f007:**
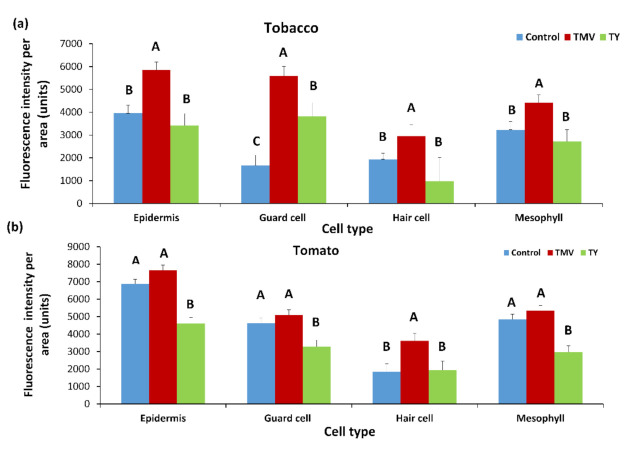
Fluorescence intensity of PDLP5-GFP protein in various cells after TYLCV and TMV infection in leaves of tobacco and tomato plants. Tobacco (**a**) and tomato (**b**) plants were stably transformed with 35S::PDLP5-GFP and were viewed in confocal microscopy 30 days after inoculation with TYLCV (TY) or TMV (TMV) or mock-inoculated (control). (**a**,**b**) Fluorescence area in the GFP per µm of the cell wall was measured per cell type. In each graph, means (± SE) are shown. Different letters above the data columns mean that differences between treatments are statistically significant (*p*{f} <  0.001), and each cell type was tested separately.

**Figure 8 plants-10-01442-f008:**
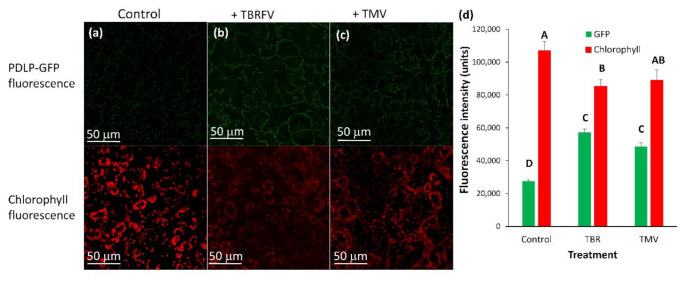
Fluorescence intensity of PDLP5-GFP protein in epidermis cells after ToBRFV and TMV infection in leaves of tomato plants (money maker). Tomato plants were stably transformed with 35S::PDLP5-GFP and were viewed in confocal microscopy 30 days after inoculation with ToBRFV (TBR) or TMV (TMV) or mock-inoculated (**control**). (**a**–**c**) pictures show GFP and chlorophyll fluorescence in various cell types; (**d**) fluorescence area of GFP or chlorophyll per µm of the cell wall was measured per treatment. Means (± SE) are shown. Different letters above the data columns mean that differences between treatments are statistically significant (*p*{f} < 0.001) in the parameter analyzed.

## Data Availability

There is no data related to this paper that is not presented.

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
