# Peer review of "Leaf Plasmodesmata Respond Differently to TMV, ToBRFV and TYLCV Infection"

_plants, 2021, doi:10.3390/plants10071442_

Round 1

Reviewer 1 Report

This article entitled "Leaf plasmodesmata respond differently to TMV, ToBRFV and TYLCV infection" describes a nice work contucted to show the respond of specific plasmodesmata located proteins to virus infections on tobacco and tomato plants. The majority of the section parts are well presented but there are some points which need to be clarified before any conclusion concerning its publication. 

  1. Lane 76 of Results sections: do the authors have any data to present about the AtPDLP5 expression in tomato (in comparisson to A. thaliana PDLP5 localization as was shown for tobacco)? Since the article is focused on differences of viruses in tobacco and tomato, in order to be more complete data should be presented for both hosts, not only for tobacco.
  2. Lines 134-138: in Fig. 6 what are the letters (A, B, BC, ..etc) above the columns stands for? It makes difficult to understand the graph. Probably they should stand as A for Epidermal cells, B for mesophyl, C for Hair cell and D for Guard cells, no? Since the columns are color-coded to my opinion there is no value to give letters to columns. Nevertheless, in tomato the graph shows that GFP fluorescence is higher in epidermis and lower in hair cells as the authors note, but in line 136 "cells show an equal and in-between fluorescence intesity..": in what do they referer (comparison to tobacco?)? can they clarify this better? The graph shows that mesophyl has higher GFP intesity than guard cells. The same for line 137 results have to rephrased and presented clearly (graph shows GFP intesity Epidermis>mesophyl>hair>gurad cells)
  3. Figure 7: again what is the meaning of these letters above columns in both pannels (might be the letters should be A for TMV, B for TYLCV and C for Control but there is a possible spelling mistake?). In line 161-162: TYLCV decreased the PDLP5-GFP fluorescence in all tomato cell types but according to the graph in Hair cells is not (or is it minor affect and not statistically different?), please correct and include this in to the results. As for the effect of TYLCV in tobacco should it be (based on the results presented in the graph): TYLCV caused a minor decrease nut not statistically different in all types of cells except for gurads cells that showed a signifiacant increase in PDLP5-GFP. Although someone could say that in Hair cell types the decrease is higher compared to the decrease presented for meshophyl and epidermis (which shown more or less equal).
  4. Figure 8: letters above columns again. This figure presents only data for one cell type but the authors mentioned that both tobamoviruses have similar affect/phenotype. Are there any data (even as data not shown) for the other types of cells in order this claim about similar mechanism of tobamoviruses to be more concrete?
  5. authors present in the text (line 252, 271) and also SFig 1, PCR data but there is no information given about the methodology even as a refferencce (for gene, primeres etc). Please include this part in Methods section.
  6. The above data have to be also presented in the discussion part (lines 211-219) as the authors present a general increase or descrease of PDLP5-GFP intensity but this is not the case for some cell types in both hosts/viruses. 
  7. Do the authors have any information or hypothesis why in some cell types the affect (increase or decrease) is higher in each virus/host combination? 
  8. Will be nice to complete this nice work if there was an experiment studying the physical interaction of MPs with this PDLP5 (if any) by BiFC, Yeast two hybrd, pull down co-precipitation etc, or to study a possible role of the protein to virus infection by VIGS. Do the authors have this as a future designed experiments, will be nice to note at least this since the specific role of the virus affect to the protein accumulation is not described

Some minor changes

  • since you mentione the abbreviation of Movement Proteins (MPs) in line 53 after you have to refer with this (eg. lines 54, 64, 197 etc)
  • line 65: "shown"
  • line 77: in AtPDLP5, please check journal guidelines if the At has to be in italics or not and apply the change in all text (eg lines 85, 87, 89 etc)
  • line 79: italics for Arabidopis
  • line 80: remove bold from the figure title
  • line 125: in figure 5 title, since the figure includes also data from tomato host is better to include this in the general title in the beginning of the legend as is mentioned "...cell wall of all tobacco cells.", but E image shows data from tomato.
  • line 145: "We examined the effect of very different plant viruses on the recruitment..."
  • line 148-150: this has to be rephrased, I suppose TMV has to be deleted and keep also only the abbreviation of ToBRFV since the full name was given before in line 42. Also if the data are only based in epidermis cells as shown in Figure 8, this has to be mentioned also in the text not only in ffigure legend.
  • line 176: "Here we showed no plasmodesmata..."
  • line 229: "The Islaeli TYLCV isolate (GenBank..."
  • line 233: "...., and ToBRFV was propagated in the tomato line..."
  • line 243: "Plants were inoculated with TYLCV..."
  • line 244 and 247: add "for" befor the "a 48-h acquisition"
  • line 257: "of tomato and/or tobacco test plants..." and/or has to be included since ToBRFV was used only for tomato plants and only TMV was used for both host inoculations.
  • line 270: please rephrase it for TMV as in this form it means that for both  TMV-tobacco and TYLCV-tobacco PCR analysis was done for verification.
  • also I noticed that the title is different in the platform system f the journal and the PDF file. Please check.

Author Response

Reviewer #1

This article entitled "Leaf plasmodesmata respond differently to TMV, ToBRFV and TYLCV infection" describes a nice work contucted to show the respond of specific plasmodesmata located proteins to virus infections on tobacco and tomato plants. The majority of the section parts are well presented but there are some points which need to be clarified before any conclusion concerning its publication. 

We thank the reviewer for the kind word.

  1. Lane 76 of Results sections: do the authors have any data to present about the AtPDLP5 expression in tomato (in comparisson to A. thaliana PDLP5 localization as was shown for tobacco)? Since the article is focused on differences of viruses in tobacco and tomato, in order to be more complete data should be presented for both hosts, not only for tobacco.

We thought it was redundant to show the tomato to save space. As suggested by the reviewer, we add the tomato pictures showing that the expression pattern of AtPDLP5-GFP in tomato is the same as tobacco.

  1. Lines 134-138: in Fig. 6 what are the letters (A, B, BC, ..etc) above the columns stands for? It makes difficult to understand the graph. Probably they should stand as A for Epidermal cells, B for mesophyl, C for Hair cell and D for Guard cells, no? Since the columns are color-coded to my opinion there is no value to give letters to columns. Nevertheless, in tomato the graph shows that GFP fluorescence is higher in epidermis and lower in hair cells as the authors note, but in line 136 "cells show an equal and in-between fluorescence intesity..": in what do they referer (comparison to tobacco?)? can they clarify this better? The graph shows that mesophyl has higher GFP intesity than guard cells. The same for line 137 results have to rephrased and presented clearly (graph shows GFP intesity Epidermis>mesophyl>hair>gurad cells)

Different letters above the data columns depict statistically significant differences between genotypes or treatments (p{f} < 0.001). I added this sentence to each figure with statistical analysis. 

The different cell types are color-coded; I increased the font and size to make it more prominent.

The sentence in line 136 is reparsed as follows: “In tomato, fluorescence is slightest in hair cells, while guard cells and mesophyll cells show an equal and in-between fluorescence intensity compared to tomato epidermis cells (Fig. 6).  In tobacco, PDLP5-GFP is evenly present in hair and  mesophyll cells, while gourd cells show the lowest GFP intensity, all compared to epidermis cells that show the highest GFP intensity (Fig. 6).”

  1. Figure 7: again what is the meaning of these letters above columns in both pannels (might be the letters should be A for TMV, B for TYLCV and C for Control but there is a possible spelling mistake?). In line 161-162: TYLCV decreased the PDLP5-GFP fluorescence in all tomato cell types but according to the graph in Hair cells is not (or is it minor affect and not statistically different?), please correct and include this in to the results. As for the effect of TYLCV in tobacco should it be (based on the results presented in the graph): TYLCV caused a minor decrease nut not statistically different in all types of cells except for gurads cells that showed a signifiacant increase in PDLP5-GFP. Although someone could say that in Hair cell types the decrease is higher compared to the decrease presented for meshophyl and epidermis (which shown more or less equal).

Different letters above the data columns depict statistically significant differences between genotypes or treatments (p{f} < 0.001). I added this sentence to each figure with statistical analysis. 

In the results line, 161 and on was changed as follows: “TYLCV caused a decrease in PDLP5-GFP recruitment in tomato cells except for hair cells of the tomato where TYLCV did not affect (Fig. 7B). TYLCV did not affect PDLP5-GFP recruitment in tobacco cells except for guard cells that showed a significant increase in PDLP5-GFP, unlike other leaf cells (Fig. 7A)”.

In tobacco, there was no statistical in PDLP5-GFP recruitment between the cell types after infection with TYLCV except for guard cells, where we observed increased fluorescence.  We measured hundreds of cells, and although there is a slight decrease, it is not statistically valid.

  1. Figure 8: letters above columns again. This figure presents only data for one cell type but the authors mentioned that both tobamoviruses have similar affect/phenotype. Are there any data (even as data not shown) for the other types of cells in order this claim about similar mechanism of tobamoviruses to be more concrete?

Different letters above the data columns depict statistically significant differences between genotypes or treatments (p{f} < 0.001). I added this sentence to each figure with statistical analysis.

As we have seen that epidermis cells are good representatives of the viral effect, we did not show other cell types as it would make a very cluterd graph. So instead, we added a picture with other cells for the reader's impression of the increased fluorescence of other cell types. The sentence was amended to “ToBRFV treated tomato plants show a similar phenotype as TMV treated tomato in both chlorophyll fluorescence decreases and PDLP5-GFP increases in epidermis cells (Fig. 8); other cell types show the same qualitative response as epidermis cells,  indicating that different tobamoviruses induce a similar effect on the plasmodesmata.”

  1. authors present in the text (line 252, 271) and also SFig 1, PCR data but there is no information given about the methodology even as a refferencce (for gene, primeres etc). Please include this part in Methods section.

The protocol was added to Methods, and we thank the reviewer for noticing the omission.

  1. The above data have to be also presented in the discussion part (lines 211-219) as the authors present a general increase or descrease of PDLP5-GFP intensity but this is not the case for some cell types in both hosts/viruses. 

We addressed this point in the discussion as follows:

It seems that TMV affects tomato and tobacco similarly by mobilization of plasmodesmata located proteins such as PDLP5-GFP to the plasmalemma and ER membranes that are part of the plasmodesmata. However, in tomato guard, epidermis, and mesophyll cells, the PDLP5-GFP increase was not significantly different, although higher than the control. This observation could be explained by the minor effect of TMV on the tomato line we used that showed minor symptoms of TMV.  ToBRFV that belongs to the same

 virus genus as TMV (Tobamoviruses), has a similar but more severe effect on both chlorophyll and GFP fluorescence in epidermis cells, pointing to the general mechanism which tobamoviruses affect plant cells.  

  1. Do the authors have any information or hypothesis why in some cell types the affect (increase or decrease) is higher in each virus/host combination? 

The PDLP5-GFP increase in fluorescence upon TMV infection and the fact that PDLP5-GFP fluorescence in tomato leaf hair cells does not respond to TYLCV infection can be explained by large plasmodesmata fields present at the base of the hair or trichome cells. The high-density plasmodesmata field at the base of the hair or trichome cells may have different properties than regular PD pits.

  1. Will be nice to complete this nice work if there was an experiment studying the physical interaction of MPs with this PDLP5 (if any) by BiFC, Yeast two hybrd, pull down co-precipitation etc, or to study a possible role of the protein to virus infection by VIGS. Do the authors have this as a future designed experiments, will be nice to note at least this since the specific role of the virus affect to the protein accumulation is not described

We agree with the reviewer, and we are trying to obtain clones that we can use for this study.

Some minor changes

  • since you mentione the abbreviation of Movement Proteins (MPs) in line 53 after you have to refer with this (eg. lines 54, 64, 197 etc)

The manuscript was amended as requested.

  • line 65: "shown"

I could not find the word

  • line 77: in AtPDLP5, please check journal guidelines if the At has to be in italics or not and apply the change in all text (eg lines 85, 87, 89 etc)

The refernce to PDLP5-GFP in the manuscript is to the protein thus, it is not in italics.

  • line 79: italics for Arabidopis

The manuscript was amended as requested.

  • line 80: remove bold from the figure title

The manuscript was amended as requested.

  • line 125: in figure 5 title, since the figure includes also data from tomato host is better to include this in the general title in the beginning of the legend as is mentioned "...cell wall of all tobacco cells.", but E image shows data from tomato.
  • The figure legend was amended as requested, both picture E and F show tomato tissue.
  • line 145: "We examined the effect of very different plant viruses on the recruitment..."
  • The text was amended
  • line 148-150: this has to be rephrased, I suppose TMV has to be deleted and keep also only the abbreviation of ToBRFV since the full name was given before in line 42. Also if the data are only based in epidermis cells as shown in Figure 8, this has to be mentioned also in the text not only in ffigure legend.
  • The text was amended
  • line 176: "Here we showed no plasmodesmata..."
  • The text was amended.
  • line 229: "The Islaeli TYLCV isolate (GenBank..."
  • The text was amended
  • line 233: "...., and ToBRFV was propagated in the tomato line..."
  • The text was amended
  • line 243: "Plants were inoculated with TYLCV..."
  • The text was amended
  • line 244 and 247: add "for" befor the "a 48-h acquisition"
  • The text was amended
  • line 257: "of tomato and/or tobacco test plants..." and/or has to be included since ToBRFV was used only for tomato plants and only TMV was used for both host inoculations.
  • The text was amended
  • line 270: please rephrase it for TMV as in this form it means that for both  TMV-tobacco and TYLCV-tobacco PCR analysis was done for verification.
  • The text was amended.
  • also I noticed that the title is different in the platform system f the journal and the PDF file. Please check.

Corrected

Reviewer 2 Report

The manuscript's authors entitled "Leaf Plasmodesmata Respond Differently to TMV and TYLCV 2 Infection" presents data on the localization of Arabidopsis thaliana Plasmodesmata-Located Protein 5 (AthPDLP5) expressed in transgenic tobacco and tomato plants. The authors have shown that constitutive expression of AthPDLP5 results in a similar to A. thaliana pattern of protein localization in cell walls of these two plant species. In the dehydrated seeds, AthPDLP5 accumulated in stress granules and was not present in cell walls. However, after the imbibition of seeds, the protein was detected mainly in cell walls. Furthermore, while AthPDLP5 localized in walls of all types of cells, they differed in the relative concentration of the protein, with the epidermis having its highest load. Infection with tobamoviruses (TMV, ToBRFV) increased the concentration of the AthPDLP5 in walls of all these cell types. Opposite to this, begomovirus (TYLCV) negatively impacted the protein concentration except for tobacco guard cells, where it had a similar stimulative impact as tobamoviruses. These differences may reflect that tobamoviruses harbor the movement protein (MP) while begomoviruses lack this protein. MPs are known for increasing the exclusion limit of plasmodesmata through interaction with different plasmodesmal proteins, including PDLP5. These data indicate that these two different groups of viruses use different mechanisms to move through the plasmodesmata.

While these results are interesting, in my personal opinion, they should be confirmed by a different approach to prove that the observed localization of the PDLP5 protein reflects what is happening with its native tobacco and tomato counterparts. That could be achieved by immunolocalization and colocalization of both native and A. thaliana PDLP5 with known cell wall markers. The authors should perform the same kind of control to quantify the differences in PDL5 relative concentration in response to viral infection. Also, the authors should investigate interactions in vitro and in vivo between viral proteins and PDLP to propose the molecular model of these interactions and clarify the differences in the mode of movement through plasmodesmata for different groups of viruses.
Therefore I recommend rejection of the manuscript in the present form.

Minor comments
Lanes 16-21 – the authors claim they have investigated two viruses, and next, they indicate that ToBRFV was also researched. This makes the abstract unclear. Clarify this section indicating that study used as a model of viruses harboring movement protein two members of tobamoviruses (names) and as a model of viruses deprived movement protein one member of begomoviruses (name). Also, write the full name of ToBRFV before using an abbreviation.

Lanes 44-45 - correct to "which is used" (you used one particular begomovirus in your study).

Fig. 3 and 5. Correct the symbol of micrometers. Check the other figures as well.

Author Response

Reviewer #2

Comments and Suggestions for Authors

The manuscript's authors entitled "Leaf Plasmodesmata Respond Differently to TMV and TYLCV 2 Infection" presents data on the localization of Arabidopsis thaliana Plasmodesmata-Located Protein 5 (AthPDLP5) expressed in transgenic tobacco and tomato plants. The authors have shown that constitutive expression of AthPDLP5 results in a similar to A. thaliana pattern of protein localization in cell walls of these two plant species. In the dehydrated seeds, AthPDLP5 accumulated in stress granules and was not present in cell walls. However, after the imbibition of seeds, the protein was detected mainly in cell walls. Furthermore, while AthPDLP5 localized in walls of all types of cells, they differed in the relative concentration of the protein, with the epidermis having its highest load. Infection with tobamoviruses (TMV, ToBRFV) increased the concentration of the AthPDLP5 in walls of all these cell types. Opposite to this, begomovirus (TYLCV) negatively impacted the protein concentration except for tobacco guard cells, where it had a similar stimulative impact as tobamoviruses. These differences may reflect that tobamoviruses harbor the movement protein (MP) while begomoviruses lack this protein. MPs are known for increasing the exclusion limit of plasmodesmata through interaction with different plasmodesmal proteins, including PDLP5. These data indicate that these two different groups of viruses use different mechanisms to move through the plasmodesmata.

  1. While these results are interesting, in my personal opinion, they should be confirmed by a different approach to prove that the observed localization of the PDLP5 protein reflects what is happening with its native tobacco and tomato counterparts. That could be achieved by immunolocalization and colocalization of both native and A. thaliana PDLP5 with known cell wall markers. The authors should perform the same kind of control to quantify the differences in PDL5 relative concentration in response to viral infection. Also, the authors should investigate interactions in vitro and in vivo between viral proteins and PDLP to propose the molecular model of these interactions and clarify the differences in the mode of movement through plasmodesmata for different groups of viruses.
    Therefore I recommend rejection of the manuscript in the present form.

We agree with the reviewer that a study on the native tobacco and tomato PDLP5 or other PDLP proteins or their interaction with MPs from various plant viruses is fascinating. Nevertheless, we could not obtain these cloned proteins readily, and doing fundamental research like that will take years.

Minor comments

  1. Lanes 16-21 – the authors claim they have investigated two viruses, and next, they indicate that ToBRFV was also researched. This makes the abstract unclear. Clarify this section indicating that study used as a model of viruses harboring movement protein two members of tobamoviruses (names) and as a model of viruses deprived movement protein one member of begomoviruses (name). Also, write the full name of ToBRFV before using an abbreviation.

The text was amended as suggested.

  1. Lanes 44-45 - correct to "which is used" (you used one particular begomovirus in your study).

The text was amended as suggested

  1. 3 and 5. Correct the symbol of micrometers. Check the other figures as well.

The figures were amended as suggested by the reviewer with bold maks and type for the micrometers and the scale for clarity.

Round 2

Reviewer 1 Report

  In the revised version the authors replied and followed the majority of the points suggested.

Nevertheless I would like the authors to make clearer some points

  • i am not convinced with the sentence the authors claim referred to Figure 6 about PDLP5-GFP intensity in tomato. According to the graph guard cells to my opinion can not be considerred as equal intensity with epidermis cells. Moreover, the figure regend has to be more precise of what exactly letters above columns stands for. Still I can not understand "Different letters above the data columns depict
    statistically significant differences between genotypes or treatments" Genotypes: the authors mean tobacco and tomato? or different genotypes of tobacco/tomato were used? In M&M I understood that Moneymarker (tomato) and SR1 (tobacco) were used. Treatments: thwy were 2 as reffered to M&M section?
  • the authors do not point any future design experiments. As this work is based to transgenic plants expressing the PDLP5 from arabidopsis and the authors did not at least made any complementary re-confirmation experiments about the tobacco and tomato PDLP5 orthologs by TEM imunogold labelling etc or a potential western blot to show the increase/decrease of the protein accumulation during viral infection of tomato/tobacco they should refer that on going research is in proggress to study the bilogical role of possible virus-PDLP5 interactions. 

Author Response

In the revised version the authors replied and followed the majority of the points suggested.

Nevertheless I would like the authors to make clearer some points

  • i am not convinced with the sentence the authors claim referred to Figure 6 about PDLP5-GFP intensity in tomato. According to the graph guard cells to my opinion can not be considerred as equal intensity with epidermis cells.

We changed to text to: ". In tobacco, fluorescence is slightest in guard cells, while hair cells and mesophyll cells show an equal and in-between fluorescence intensity compared to tobacco epidermis cells (Fig. 6).  In tomato leaf, PDLP5-GFP shows the lowest GFP intensity in hair and guard cells (Fig. 6).  , while mesophyll cells in tomato leaf show an in-between GFP intensity and epidermis cells show the highest GFP intensity (Fig. 6). "

  • Moreover, the figure regend has to be more precise of what exactly letters above columns stands for. Still I can not understand "Different letters above the data columns depict statistically significant differences between genotypes or treatments" Genotypes: the authors mean tobacco and tomato? or different genotypes of tobacco/tomato were used? In M&M I understood that Moneymarker (tomato) and SR1 (tobacco) were used. Treatments: thwy were 2 as reffered to M&M section?

We changed to explanation in figures 6 to 8 to the following: "Different letters above the data columns mean that differences between treatments are statistically significant (p{f} < 0.001)."

  • the authors do not point any future design experiments. As this work is based to transgenic plants expressing the PDLP5 from arabidopsis and the authors did not at least made any complementary re-confirmation experiments about the tobacco and tomato PDLP5 orthologs by TEM imunogold labelling etc or a potential western blot to show the increase/decrease of the protein accumulation during viral infection of tomato/tobacco they should refer that on going research is in proggress to study the bilogical role of possible virus-PDLP5 interactions. 

The following  sentences were added: "We are trying to transform tomato plants with MP from ToBRFV and TYLCV to analyze the interaction with PDLP5-GFP by crossing the two transgenic plants. "

We are currently also working on the distribution of PDLP5-GFP during shoot regeneration in tomato cotyledons explants and the response PDLP5-GFP  in these cotyledons to phytohormones. 

Reviewer 2 Report

After reading the author's response, I concord that obtaining antibodies to additional proteins may take years and, therefore, be published in the following paper. Moreover, I noticed that the authors had revised all other comments. Therefore I recommend publishing as it is.

Author Response

We are thankful to the reviewer for the recommendation.

Round 3

Reviewer 1 Report

The authors followed all the points suggested in the revised article (round 2). It can be accepted as it is.